# Genotypes Distribution of Epstein–Barr Virus among Lymphoma Patients in Ethiopia

**DOI:** 10.3390/ijms241813891

**Published:** 2023-09-09

**Authors:** Seifegebriel Teshome, Elshafa Hassan Ahmed, Kidist Zealiyas, Abdulaziz Abubeker, Fisihatsion Tadesse, Christoph Weigel, Robert A. Baiocchi, Tamrat Abebe

**Affiliations:** 1Department of Microbiology, Immunology and Parasitology, Addis Ababa University, Addis Ababa 9086, Ethiopia; seifegebriel.teshome@aau.edu.et; 2Division of Hematology, Department of Internal Medicine, College of Medicine, The Ohio State University, Columbus, OH 43210, USA; elshafa.ahmed@osumc.edu (E.H.A.); christoph.weigel@osumc.edu (C.W.); 3Ethiopian Public Health Institute (EPHI), Addis Ababa 1242, Ethiopia; kzealiyas@gmail.com; 4Aklilu Lemma Institute of Pathobiology, Addis Ababa University, Addis Ababa 1176, Ethiopia; 5Department of Internal Medicine, Addis Ababa University, Addis Ababa 9086, Ethiopia; abdulazizas88@gmail.com (A.A.); fishtadat@yahoo.com (F.T.); 6Comprehensive Cancer Center, The James Cancer Hospital and Solove Research Institute, The Ohio State University, Columbus, OH 43210, USA

**Keywords:** Epstein–Barr virus, genotypes, lymphoma, *EBNA1*, *EBNA3C*, Ethiopia

## Abstract

Epstein–Barr virus (EBV) is an oncogenic herpes virus associated with several human malignancies. Two main EBV genotypes (type 1 and type 2) distinguished by the differences in EBV nuclear antigens are known. Geographic variability in these genetic differences has been observed in the incidence of some EBV-related tumors. Here, we investigated the genetic variation of EBV in lymphoma specimens collected in Ethiopia. A total of 207 DNA samples were used for EBV detection and typing, and *EBNA1* and *EBNA3C* genes were used to detect and subtype the EBV genome, respectively. EBV genotype 1 was detected in 52.2% of lymphoma patients. EBV genotype 2 was detected in 38.2% of the lymphoma patients, and 9.7% were coinfected by both EBV genotypes. Overall, 52.8% of the Hodgkin’s lymphoma (HL) patients and 51.8% of non-Hodgkin’s lymphoma (NHL) patients showed the presence of genotype 1. Meanwhile, 42.8% and 2.3% of HL patients and 35.8% and 12.4% of NHL patients showed EBV genotype 2 and both genotypes, respectively. Significant associations between the age groups and EBV genotypes were observed (*p* = 0.027). However, no significant association was seen between EBV genotypes and other sociodemographic and clinical characteristics. This study showed that the distribution of EBV genotype 1 was higher in Ethiopian lymphoma patients.

## 1. Introduction

Epstein–Barr virus (EBV), also called human herpesvirus 4, was first identified in 1964 by Epstein’s group in a cell line derived from Burkitt lymphoma [1]. EBV infects both lymphoid and epithelial cells and causes different EBV-associated cancers. The origins of all of these tumors are arising from specific stages in the EBV life cycle and appear to be associated with disturbances of the immune system [2].

EBV is associated with the development of several malignancies. These malignancies are known as those in which viral Deoxyribonucleic acid (DNA) or Ribonuclease acid (RNA) and/or viral gene expression can be demonstrated in the tumor tissues [3]. Associated diseases include Hodgkin’s disease (HD), Burkitt’s lymphoma (BL), diffuse large B cell lymphoma (DLBCL) and post-transplant lymphoproliferative disorders (PTLDs), Oral hairy leukoplakia (OHL), Nasopharyngeal carcinoma (NPC), and Undifferentiated gastric carcinoma [4].

These EBV-associated malignancies are mainly associated with latent viral infections, and this latency is controlled by various viral genetic and epigenetic regulatory mechanisms [5]. EBV expresses several of its genes during latency. EBV nuclear antigens (EBNAs) are encoded during latency from several alternatively spliced primary transcripts to form *EBNA1*, *EBNA2*, *EBNA3A, EBNA3B, EBNA3C*, and *EBNA* leader protein (*EBNA-LP*). The latent membrane proteins (LMPs), *LMP1*, *LMP2A*, and *LMP2B*, are expressed from individual promoters. EBV also expresses non-coding RNAs such as EBV-encoded small RNAs (*EBER*)*1* and *EBER2*, respectively, and many viral microRNAs (miRNAs) [6,7]. Currently, a total of 44 EBV-encoded miRNAs are known [8].

Lytic EBV state has also been shown to contribute toward EBV-associated lymphomagenesis in preclinical in vivo models, virus strains with increased lytic EBV replication are enriched in EBV-associated malignancies, and plasma viral loads correlate with some of these diseases [9]. Lytic infection contributes to EBV oncogenicity by raising the number of latently infected B cells, which increases the burden of latently infected B cells and potential immortalization/transformation. Lytic phase can also contribute toward oncogenesis by promoting the expression of pro-inflammatory cytokines, growth factors, and distinct cellular signaling pathways that promote cell proliferation and genomic instability [10,11]. This has been observed in EBV associated BL occurring due to the coinfection of EBV with malaria infections [12].

EBV is categorized into two types based on the genetic difference of *EBNAs* (*EBNA2*, *-3A*, -*3B*, and *-3C*). These two genotypes are called EBV type 1 (e.g., B95-8) and EBV type 2 (e.g., AG876) or type A and B, respectively [13]. *EBNA2* has only 70% homology at the gene level and 54% at the protein level when comparing EBV type 1 vs. type 2 [14]. Differences between EBV type 1 and type 2 genotypes were observed from EBV isolated from North America and from central Africa from infectious mononucleosis and Burkitt Lymphoma patients, respectively [15,16]. In addition to geographic variability, the two genotypes also display differences in transforming capacity, where EBV genotype 1 has higher efficiency in immortalizing B cells than genotype 1 [17,18]. However, recent studies showed that EBV type 2 were more effective in infecting T cells [19,20].

EBV detection among different EBV-associated malignancies varies depending on geographic regions, age, and sex [21]. Geographically, EBV type 1 is prevalent in populations from Europe, America, China, and South Asia. In contrast, type 2 is less prevalent in these populations and is more often observed in African and Papua New Guinean populations, where immunocompromised patients are more susceptible to acquiring both types [22]. These strains have different abilities in transforming B cells, and EBV type 1 transforms B cells more efficiently than type 2 [23,24,25].

Sequence variations of the two genotypes are mainly performed by targeting *EBNA2* or *EBNA3* genes of the virus [16]. Different molecular techniques are used to differentiate EBV genotypes across various geographic locations, study groups, and EBV-associated diseases [26,27,28]. Detection of *EBNA2* and *EBNA3C* using nested [29] and conventional polymerase chain reactions (PCR) is widely used. Targeting the *EBNA3C* gene sequence is the preferred method to characterize the EBV genotypes among different population groups [30,31]. 

A recent advancement in sequencing technology has increased our understanding of the genetic variations that exist across different populations globally [32]. Sanger sequencing techniques [33] and next-generation sequencing platforms are crucial in assessing the genotype and strain variation of EBV [23,34,35]. The two genotypes have a higher degree of nucleotide polymorphism within multiple viral genes. Of the nine latent EBV genes, LMP1 has a greater degree of polymorphism. Variants in LMP-1 have been classified into seven main groups: B95-8, Alaskan, China 1, China 2, Med+, Med−, and NC [36,37]. EBV polymorphisms within different open reading frame (ORF) loci have also been shown to be associated with various malignancies [38,39]. Polymorphisms in EBNA 3B is associated with a variety of EBV associated malignancies [40], while polymorphisms in the EBER locus are associated with NPC [41]. 

Understanding genetic variation in different geographic locations is crucial for disease monitoring and evaluation. EBV type variation in different sites among different malignancies has been reported [42,43]. However, studies showing the distribution of the EBV genotypes in sub-Saharan Africa are very limited. Especially in Ethiopia, studies concerning the burden of EBV and its genotype distribution among different EBV-associated hematologic malignancies have never been performed. Therefore, this study aimed to determine the genetic variability of EBV among lymphoma patients. 

## 2. Results

### 2.1. Sociodemographic and Clinical Characteristics

A total of *n* = 207 DNA samples were collected from FFPE, PBMCs, and LMNCs for EBV detection and subtyping. A total of 63 (30.4%) DNA samples were isolated from FFPE lymphoma blocks, 118 (57%) from PBMCs, and 26 (12.6%) of the DNA samples were extracted from LMNCs. The majority of the study participants were male patients (*n* = 135, 65.2%). The study participants age ranged from 3 to 80 years, with a mean age of 38.91 ± 16.94 SD years. The overall clinical and sociodemographic data of the study participants are listed in Table 1.

Out of the 137 non-Hodgkin’s lymphoma (NHL) study participants, the majority (*n* = 35, 25.5%) were small lymphocytic lymphoma (SLL) patients, followed by diffuse large B cell lymphoma patients (*n* = 33, 24.1%). The distribution of the different lymphoma types in the study participants showed in Figure 1. Male patients comprised 65.7% (*n* = 46) and 65% (*n* = 89) of HL and NHL patients, respectively. The predominant age group in HL patients was the younger population. A total of 32 (45.7%) of HL patients were categorized between the age of 20 and 40. However, in NHL patients, the majority (*n* = 63, 46.3%) of patients were categorized under the age group between 40 and 60 years. Among NHL patients, HIV prevalence was higher (*n* = 9, 27.3%) in DLBCL patients than in the other NHL types.

### 2.2. EBV Detection

EBV detection and quantification were performed using *EBNA1*. All samples had a detectable viral load ranging between 10^2^ EBV copies/mL to 10^9^ EBV copies/mL with a mean count of 2.8 × 10^9^ EBV copies/mL. The large majority (*n* = 147, 71%) of the lymphoma patients had a high EBV viral load (>10,000 EBV copy number/ml), whereas 20.8% (*n* = 43) and 8.2% (*n* = 17) of them had a low (between 5000–10,000 EBV copy number/mL) and very low (<5000 EBV copy number/mL) EBV viral load, respectively. Out of these (*n* = 147) patients with higher EBV viral load, the majority were male (*n* = 97, 66%), NHL (*n* = 97, 66%), and between the age of 20 and 40 (*n* = 60, 41%). Of the 207 study participants, 28 were HIV positive, and of these, 85% (*n* = 24) had a high EBV viral load (>10,000). Of the NHL patients with a higher EBV viral load (*n* = 97), 25 patients (25.8%) were DLBCL patients, followed by SLL (*n* = 24, 24.7%) (Figure 2). We did not observe any significant association between sociodemographic and clinical variables with EBV viral load (high, low, and very low).

### 2.3. EBNA3C Typing

EBV DNA was detected in all (*n* = 207) lymphoma cases, and then EBV type was identified using *EBNA3C*. Out of the 207 lymphoma patients, 52.2% (*n* = 108) had EBV genotype 1, while 79 (38.2%) had EBV genotype 2 and 20 (9.7%) were coinfected with both genotypes (Figure 3).

Next, we assessed if the type of EBV infection varied per sex, HIV status, and type of lymphoma. Our results showed no difference between males and females in the type of EBV genotype infection. However, a significant association between the age groups and EBV genotypes was observed (*p* = 0.027). EBV genotype 1 was higher in age groups less than 20 years (*n* = 20/32, 62.5%), while EBV genotype 2 was higher within the age group of 21 to 40 (*n* = 37/79, 46.8%). Among HIV-positive and EBV-positive lymphoma patients (*n* = 28), EBV genotype 1, genotype 2, and coinfection was detected in 13 (46.4%), 12 (42.8%), and 3 (10.7%) of HIV-positive lymphoma patients, respectively (Table 2).

Of the *n* = 137 NHL types, SLL (*n* = 35, 25.5%) and DLBCL (*n* = 33, 24.1%) comprise most of the study population. From *n* = 35 SLL cases, EBV genotype 1 was detected in *n* = 16 (45.7%), EBV genotype 2 was detected in *n* = 12 (34.3%), and *n* = 7 (20%) represented coinfection with both genotypes. Of the *n* = 33 DLBCL cases, EBV genotype 1, EBV genotype 2, and both types were detected in *n* = 18 (54.5%), *n* = 11 (33.3%), and *n* = 4 (12%) patients, respectively. There was no significant association between the different NHL types and the distribution of EBV genotypes (*p* value = 0.92). The overall genotype distribution of EBV among different NHL types is indicated in Figure 4a. Of the samples with high EBV viral load (*n* = 147), EBV type 1 was detected in *n* = 73 (49.7%), EBV type 2 in *n* = 61 (41.5%), and both types in *n* = 13 (8.8%). There was no significant association between EBV viral load and EBV genotypes (*p* value = 0.57) (Figure 4b).

## 3. Discussion

The analysis of the EBV genetic variability is essential to understand the disease pathogenesis and the overall impact of the EBV in lymphoma. In our study, we provided the first evidence-based data that showed the genotype distribution of EBV among lymphoma patients in Ethiopia. We characterized EBV genotypes based on the *EBNA3C* gene sequences.

EBV genotyping can be made by targeting various genes such as *EBNA2* and *EBNA3C* genes. Variation in the EBV *EBNA3C* gene has been reported to classify EBV types [42]. Hence, we determined the EBV genotype based on the proposed suggestion by Sample et al. that determined the two genotypes had a significant divergence at four genetic loci and maintained type-characteristic differences at each locus [16]. Based on *EBNA3C* gene typing, we determined EBV genotype 1 in 52.2%, EBV type 2 in 38.2%, and type 1 and 2 coinfections in 9.7% of the lymphoma patients. 

The majority of studies from other parts of the world showed a higher predominance of EBV type 1, while EBV genotype 2 was predominant in sub-Saharan Africa and Papua New Guinea [22,44]. EBV type 1 was isolated from 71.1% of the study populations in Brazil by Monteiro TAF et al. [45]. In addition, EBV type 1 was detected in other studies conducted on different lymphoma types in Pakistan by Salahuddin S et al. (90.7%) [29], 91.2% from Iran on hematologic malignancies by Tabibzadeh et al. [46], 76.3% from China on NPC by Cui Y et al. [47], and 72.5% from Qatar on healthy individuals by Smatti M et al. [48]. All these findings indicated a higher EBV type 1 prevalence compared to our study (52.2%), which is also seen in other countries. This difference could account for different factors in our study, where we used the *EBNA3C* gene to determine EBV genotypes. In their review, Smatti et al. [49] showed that EBNA3 shows less difference than EBNA2. Besides, we are reporting data from only 207 patients, which may not represent the lymphoma patients in Ethiopia or the population in sub–Saharan Africa.

Comparable findings with our result were also observed from some studies conducted in the USA from samples collected from healthy individuals by Sixbey JW et al., who identified EBV type 1, type 2, and both genotypes in 50%, 41%, and 9% of the study participants, respectively [50]. Another study on lymphoma patients revealed a 56% prevalence of EBV type 1, 13% with type 2, and 31% with dual viral infections [51]. A study conducted in Brazil on HIV patients showed a 47.37% EBV type 1 prevalence [52].

Our study revealed that EBV type 2 infection was 38.2%, and such high EBV genotype 2 predominance was observed in some sub-Saharan African countries. A study conducted in Ghana showed a 52% EBV type 2 prevalence [53]. Other studies in other regions also revealed a high predominance of EBV type 2. A study conducted in China on lymphoma patients assessed a 98.6% EBV type 2 prevalence [27]. In HL patients in Mexico, EBV type 2 was discovered in 47.6% of children and 69.2% of adults [28]. Another study in Turkey showed a 44.4% EBV type 2 prevalence [54]. These disparities in the prevalence of the EBV genotypes could be attributed to the differences in their oncogenic property, the immune status of the different study populations, and geographic and strain variations [55,56,57]. 

Comparable genotype prevalence was observed in HL and NHL patients from our study, and this was similar to a study conducted in Zambia on HL patients that identified EBV type 1 in 55.6% of cases. In comparison, 33.3% were EBV type 2, and 11.1% were type 1 and 2 coinfected cases [31]. On the contrary, the prevalence of EBV genotype 1 in our study in both HL patients was lower compared to other European studies, such as those conducted in Belgium [58] and Croatia [30]. These disparities could be mainly due to the geographic variations and immune status of the populations within each country. 

There were no significant associations between the EBV genotypes and the different lymphoma types. Similar reports were revealed in other studies where lymphoma type did not affect the distribution of EBV genotypes [29,46]. However, we found significant differences in EBV types with age groups, similar to a study conducted in Brazil [28,59]. In our study, despite the age groups, a significant association between EBV genotypes with other sociodemographic and clinical data like sex, HIV status, lymphoma types, and EBV viral load was not observed. 

Some reports suggest that coinfection with EBV type 1 and type 2 is possible in immunocompromised patients, indicating infection with EBV type 2 may be acquired during the patient’s immunocompromised state [52,60]. Higher coinfection rates with both EBV types were determined on HIV patients from studies conducted in Argentina (32.6%) [61], China (12.96%) [62], and Brazil (26.32%) [52] compared to our coinfection report (9.7%).

## 4. Materials and Methods

### 4.1. Study Design and Setting

A hospital-based cross-sectional study was conducted at Tikur Anbessa specialized hospital, Addis Ababa, Ethiopia from June 2020 to March 2022. The hospital is one of the top referral hospitals serving the entire nation in providing all forms of medical service. It is the largest of the public teaching hospitals, having more than 26 departments. Both prospective and retrospective study designs were applied to recruit the study participants. Lymphoma patients who were under follow-up at hematology clinics and willing to participate were enrolled in the study. 

### 4.2. Lymphoma Samples and Clinical Data

We have collected samples from formalin-fixed paraffin-embedded (FFPE) lymphoma tissue blocks retrospectively and blood and fresh lymph node tissue samples from lymphoma patients prospectively at Tikur Anbessa specialized hospital, Addis Ababa, Ethiopia. A total of 207 samples were collected, and out of these, 63 of them were FFPE lymphoma blocks, 118 were blood samples from lymphoma patients, and the rest 26 were lymph node biopsies. An amount of 20 mL of venous blood was collected using ACD tubes, and a block (approximately 5 mm^3^) was removed from fresh lymph node biopsy tissue. Peripheral blood mononuclear cells (PBMCs) were extracted from blood fusing Ficoll-Paque techniques, and lymphomononuclear cells (LMNCs) were isolated immediately from fresh lymph node biopsy. The isolated cells were counted, and the viability of the cells was determined using the trypan blue staining method. The study participants’ medical records were reviewed, and questionnaires were also used for prospective study participants. Sociodemographic data including age, sex, lymphoma status, and other clinical data were collected. Ethical approval was obtained from the departmental ethical approval committee, the College of Health Sciences Institutional Review Board, Addis Ababa University (CHS-IRB), and the National Research Ethical Review Committee (NRERC). The study participants gave their consent before sample collection.

### 4.3. Extraction of Genomic DNA 

Two 5 µm FFPE tissue samples were deparaffinized using the deparaffinization solution, which was ready to use (QIAGEN, Hilden, Germany). After adding the deparaffinization solution, extraction of FFPE DNA was performed using the QIAamp DSP DNA FFPE tissue kit (QIAGEN, Hilden, Germany). Extraction of genomic DNA from PBMCs and LMNCs was performed using a QIAamp mini-DNA kit according to the manufacturer’s instructions (QIAGEN, Hilden, Germany). The final elusion volume of the extracted DNA was 40 µL for FFPE DNA and 100 µL for DNA extracted from PBMCs and LMNCs. The extracted DNA’s concentration and quality were measured using both Qubit (Invitrogen, Waltham, MA, USA) and Nanodrop (Thermo Scientific, Waltham, MA, USA) spectrophotometers. The concentration of the DNA using qubit ranged from 8.6 ng/µL up to 908.4 ng/µL, and the quality of the DNA using nanodrop by measuring the absorbance at 260/280 nm value was between 1.7 and 2.2. The volume of elution to determine the quality and quantity of the extracted DNA was 2 µL.

### 4.4. EBV EBNA1 Gene Detection 

EBV detection was performed using real-time quantitative PCR. An amount of 10 ng of DNA was used as starting material for qPCR, and the assay was carried out using a Viia7 qPCR machine (Applied Biosystems, Waltham, MA, USA). The reaction was performed using a 5 µL of 2× Fast SYBR green master mix (Applied Biosystems), 0.25 µL of forward (10 µM) and 0.25 µL of reverse (10 µM) primers, 2.5 µL of distilled water, and 2 µL of 5 ng/µL DNA concentration with a total reaction volume of 10 µL. qPCR consisted of 40 reaction cycles with denaturation at 95 °C for 1 s, annealing at 60 °C for 20 s, and extension at 70 °C for 30 s. Raji cell line was used as positive control, and K562 were used as a negative control. These cell lines were obtained from the American Type Culture Collection (ATCC) and have been stored in the laboratory for a long time (ATCC number CCL-86 and CCL-243, respectively). Twelve standards for *EBNA1* and *ACTB PCR products* with different concentrations were also used to quantify the EBV viral load. The EBNA1 gene standard was obtained from *EBNA1* amplicon with accession number *EBNA1*-NC_007605.1. The stock concentration of the standard was 1.98 ng/µL, and the different concentration of the standards were obtained through a twofold dilution. The range of the *EBNA1* gene standards in copy number spanned from 2.6 × 10^6^ copies/µL to 1.2 × 10^3^ copies/µL, starting from the highest dilution factor to the lowest.

The *ACTB* gene is a housekeeping gene for normalizing the host genome DNA, and we have used actin beta obtained from Homo Sapiens with the accession number NM_001101.5. The stock concentration of the *ACTB* standard was 4.3 ng/µL, and different concentrations of the standards were obtained through a twofold serial dilution. The standards for the *ACTB* gene in copy number ranged from 8.2 × 10^5^ copies/µL to 4 × 10^2^ copies/µL, starting from the highest dilution factor to the lowest. Standard curve was determined based on the CT values of each twelve standards.

The nucleotide position for the *ACTB* gene locus on the forward primer was at the nucleotide position from 707 to 726, and for the reverse primer nucleotide position, it was from 814 to 794. The size of the amplicon fragment length was 108 bp. The qPCR was performed using a total reaction volume of 10 µL consisting of 5 µL of 2× Fast SYBR green master mix (Applied Biosystems), 0.25 µL of forward (10 µM) and 0.25 µL of reverse (10 µM) primers, and 2.5 µL of distilled water and 2 µL of the standards. The thermal profile of the reaction was as follows: denaturation at 95 °C for 1 s, annealing at 60 °C for 20 s, and extension at 70 °C for 30 s, which consists of 40 cycles. All the standards, samples and controls were run in triplicate. Each CT value was calculated and converted into EBV copies/mL to obtain the EBV viral load using the ΔΔCt method.

### 4.5. EBV Typing Using EBNA3C Gene

EBV typing was determined by using the *EBNA3C* gene with a specific primer (Table 3) designed for targeting the region of divergence between EBV type 1 (B95 coordinate 87,651–87,669) and EBV type 2 (B95-8 coordinate 87,803–87,783) [31]. These coordinates are nucleotide sequences at the reference genome that have an accession number of NC_007605.1. PCR was performed in 20 µL of total solution using 10 µL of NEBNext Q5 Hot Start HiFi PCR Master Mix (New England Biolabs, Ipswich, MA, United States), 1 µL of forward (10 µM) and 1 µL of reverse primers (10 µM), 3 µL of distilled water, and 5 µL of DNA sample with a total concentration of 50 ng. The reaction mixture was initially denatured at 98 °C for 30 s, followed by 50 cycles of denaturation at 98 °C for 10 s, annealing at 68 °C for 30 s, and extension at 72 °C for 10 s. Then, 2% agarose gel electrophoresis was performed after the PCR, and B95-8 (ATCC VR-1492) and Jijoye (ATCC CCL-87) cell lines were used as a reference for EBV type 1 and 2 genotypes, respectively. These cell lines were also purchased from ATCC and stored in the laboratory. EBV type 1 and 2 genotypes were differentiated through the amplification product of 153 bp for EBV-1 and a product of 246 bp for EBV-2. 

### 4.6. Statistical Analysis

SPSS version 26.0 was used to calculate descriptive statistics and Chi-square tests to study the association between the presence of EBV with lymphoma type as well as other sociodemographic and clinical data. *p*-values < 0.05 were considered statistically significant.

## 5. Conclusions

Wide-ranging characterization of EBV provides new insights into understanding the role of EBV in different EBV-associated malignancies. Knowing the molecular diversity of EBV in various malignancies from different populations is crucial for understanding the viral effects on host immunobiology, developing new preventive strategies, and finding new approaches for innovative vaccine development. With this study, we provide baseline information for future studies in the country on this field.

## Figures and Tables

**Figure 1 ijms-24-13891-f001:**
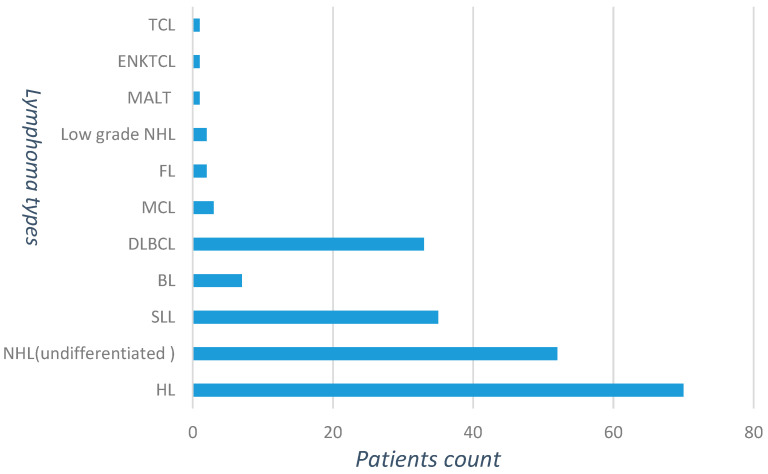
Distribution of lymphoma types among the study participants (*n* = 207). Abbreviations: DLBCL: diffuse large B cell lymphoma; BL: Burkitt lymphoma; HL: Hodgkin’s lymphoma; NHL: non-Hodgkin’s lymphoma; SLL: small lymphocytic lymphoma; FL: follicular lymphoma, MCL: mantle cell lymphoma, ENKTCL: extranodal natural killer/T-cell lymphoma. MALT: mucosa-associated lymphoid tissue lymphoma.

**Figure 2 ijms-24-13891-f002:**
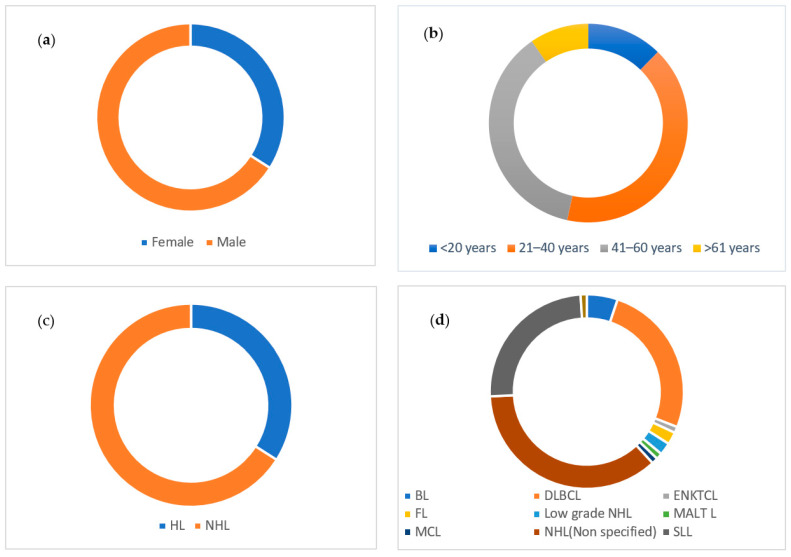
Detection of higher EBV viral load across different sociodemographic and clinical characteristics. (**a**) High EBV viral load by gender; (**b**) high EBV viral load by age group; (**c**) high EBV viral load by Lymphoma type; (**d**) high EBV viral load by NHL types. Abbreviations: DLBCL: diffuse large B cell lymphoma; BL: Burkitt lymphoma; HL: Hodgkin’s lymphoma; NHL: non-Hodgkin’s lymphoma; SLL: small lymphocytic lymphoma; FL: follicular lymphoma, MCL: mantle cell lymphoma, ENKTCL: extranodal natural killer/T-cell lymphoma.

**Figure 3 ijms-24-13891-f003:**
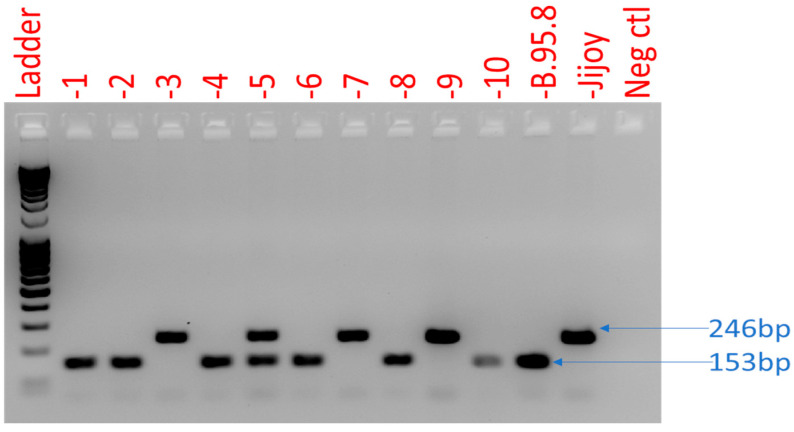
Representation of EBV genotyping using *EBNA3C* gene typing using conventional PCR. Numbers 1 to 10 refer to examples for patient sample PCRs, B95.8 refers a positive control for EBV genotype I, Jijoy refers to positive control for EBV genotype 2, and Neg Ctl refers to EBV negative cell line used as control.

**Figure 4 ijms-24-13891-f004:**
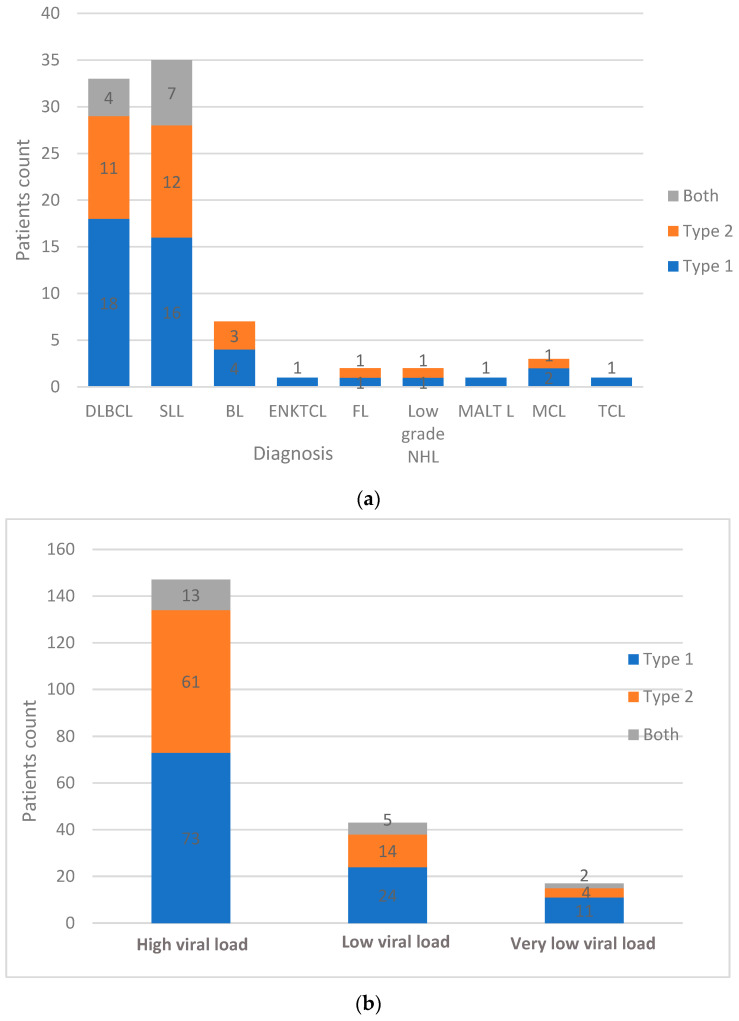
The magnitude of EBV genotypes. (**a**) EBV genotype distribution within different non-Hodgkin’s lymphoma types; (**b**) EBV viral load status vs. EBV genotypes. Abbreviations: DLBCL: diffuse large B cell lymphoma; BL: Burkitt lymphoma; NHL: non-Hodgkin’s lymphoma; SLL: small lymphocytic lymphoma; FL: follicular lymphoma, MCL: mantle cell lymphoma, ENKTCL: extranodal natural killer/T-cell lymphoma. MALT L: mucosa-associated lymphoid tissue lymphoma.

**Table 1 ijms-24-13891-t001:** Clinical and sociodemographic profile of the study participants.

Characteristics	Number	Percent
Study participants (*n* = 207)		
FFPE blocks	63	30.4
Lymphoma patients	144	69.6
Sex		
Male	135	65.2
Female	72	34.8
Age		
<20 years	32	15.5
21–40 years	79	38.2
41–60 years	75	36.2
>61 years	20	9.7
Not applicable	1	0.5
HIV status		
Positive	28	13.5
Negative	98	47.3
Unknown	81	39.2
Lymphoma types		
Hodgkin’s lymphoma	70	33.8
Non-Hodgkin’s lymphoma	137	66.2

**Table 2 ijms-24-13891-t002:** Distributions of EBV genotypes across sex, HIV status, and disease phenotype.

Characteristics	EBV1	EBV2	Coinfections	Total	*p*-Value
Age <20 years 21–40 years 41–60 years >60 years Not applicable	20 (62.5%) 39 (49.4%) 36 (48%) 12 (60%) 1	11 (34.4%) 37 (46.8%) 25 (33.3%) 6 (30%)	1 (3.1%) 3 (3.8%) 14 (18.7%) 2 (10%)	32 79 75 20 1	0.027
Sex Male Female	70 (51.8%) 38 (52.8%)	52 (38.5%) 27 (37.5%)	13(9.6%) 7(9.7%)	135 72	0.99
HIV status HIV positive HIV negative Not applicable	13 (46.4%) 45 (45.9%) 50 (61.7%)	12 (42.9%) 42 (42.9%) 25 (30.9%)	3 (10.7%) 11 (11.2%) 6 (7.4%)	28 98 81	0.169
Lymphoma type HL NHL	37 (52.9%) 71 (51.8%)	30 (42.6%) 49 (35.7%	3 (4.3%) 17 (12.4%)	70 137	0.15

**Table 3 ijms-24-13891-t003:** DNA primers used for PCR reaction.

PCR Reactions	Forward	Reverse
*EBNA1*	TCATCATCATCCGGGTCTCC	CCTACAGGGTGGAAAAATGGC
*ACTB*	TCGTGCGTGACATTAAGGAG	CAGGCAGCTCGTAGCTCTTC
*EBNA3C*	AGAAGGGGAGCGTGTGTTG	GGCTCGTTTTTGACGTCGG

## Data Availability

Available upon request.

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
