# Peer review of "Genotypes Distribution of Epstein–Barr Virus among Lymphoma Patients in Ethiopia"

_ijms, 2023, doi:10.3390/ijms241813891_

Round 1

Reviewer 1 Report

Epstein - Barr virus (EBV) is extremely ubiquitous and is consistently detected in numerous cancers, including nasopharyngeal carcinoma, subtypes of Hodgkin and non-Hodgkin lymphomas. Therefore, analysis that gives insight into the molecular polymorphism, evolution and adaptation of EBV are important.  

 Paper aimed to determine the genetic variability of EBV among lymphoma patients in Ethiopia. Undoubtedly, this studies may be valuable, as provide some new information about EBV polymorphism.

Nevertheless, I have some suggestions and comments.

 Abstract – concise and well-written, gives the most important information about paper.

 Introduction 

Mention about lytic phase of infection as different genotypes may have different proprieties in term of infection.  EBV genotype 1 is more efficient in immortalizing B cells while genotype 2 has a higher lytic ability 

 Lines 63-69 – I think that you can also include Sanger sequencing as method of EBV genotyping. Could you give more details about polymorphic nucleotides between genotypes ?

 Materials and methods

4.1. Specify how many samples from each group have you tested – FFPE – n= ? This information should be included in the materials and methods section.

 4.2.  Please, provide the protocol of deparaffinization.

 Include quality and quantity of DNA obtained after Nanodrop and Qubit measurement (even as supplementary or range).

 Have you evaluated the quality and integrity of the extracted DNA for example by PCR amplification of  internal control (for example b-globine, GAPDH – important especially in case of FFPE samples) ? What was the final volume of elution ?

4.3. – Line 234 – ten with capital letter.

Why you decided to use dye-based reaction instead of probe ? Have you verified specificity of the reaction ? As SYBR green may attach to the non-specific double stranded DNA.

 Give more details about the standards. How they were obtained ? Were they plasmids with known copy number ? What was the concentration range covered by the standards ?  

 You mentioned about ACTB PCR products and provided primers sequence for this product, but there is no explanation of this fragment. I suppose that it is b-actin, but you should clarify it and include the details about the composition of reaction mix, and thermal profile of the reaction. 

 Give more details about the targeted regions (nucleotide positions) and fragments length.

 4.4. Provide the accession numbers for the sequence you used as reference to point divergence region

 What mean coordinates ? B95 coordinate 87651-87669 and B95-8 coordinate 87803-87783 mean the fragment of the sequence to which forward and reverse primers anneal ?

 I think that you bear in mind the statement which we can also find in the paper Kafita et al. 2018

 “The sequences and positions of these primers are as follows: EBVD-F, 5'-AGAAGGGGAGCGTGT GTTGT-3' (B95-8 coordinate 87651-87670); EBVD-R, 5'-GGCTCGTTTTTGACGTCGGC-3' (B95-8 coordinate 87803-87784), which yield an amplification product of 153bp for EBV-1 and a product of 246bp for EBV-2.”

 If so, explain that coordinates point the positions of these primers in the targeted sequence. 

 Line 261 – 4.5. Statistical analysis section is inserted into the tables. Correct it.

 Results

2.1. - You should chose on way of number notation (word notation or numerical notation)– sixty-three or 63, and twenty-six or 26. Do not combine it.

 Table 1 – include column p-value, and point the statistical significance.

 Line 90 – Number of patients with DLBCL is definitely higher than 2, please correct it.

 2.2. Line 106 – Viral load is usually presented as 10n. In my opinion such notification is more clear, please  correct it.

Line 107 – Is the threshold values for  high and low viral load derived from some reference, or it was set by you ? You should also keep one notation copies/ml or particles/ml.

 Figure 2 – Why you decided to include the figure only for the high viral load ? It would be worthy to include similar figures for low and very low replication, even as supplementary materials.

You should also explain if the differences between groups (high, low, and very low replication) were statistically significant.

 2.3. – Figure 3 - Minor issue, if it is possible, try to enhance quality of the figure to reduce the red blur in some bands. You should add information concerning Jijoy and B.95.8, I mean that you should explain in caption that they are positive controls for genotypes 1 and 2. Instead of H2O, just negative control .

 I think that this section is the strongest point of the paper especially table 2, which provides some interesting information. It is pity that there is no clinical parameters, as they may have really nice clinical meaning.

 Lines 137-140 – Statistical significance ?

 Figures 4a and 4b – Improve the quality of these figures as they are impossible to trace.

Full names are not displayed in case of diagnosis, numbers are not clear. In figure B, you should precise that it concerns replication, so correct the x-axis to high replication, low-replication, and so on. Explain if there were any statistically significant differences.

 Discussion

Generally, this section is quite well-written. Authors compare their results with the EBV genotypes distribution investigated in other countries.

 Line 195 – lack of space

Lines 203-204 – This issue should be discussed more extensively. 

 Decision

Paper seems to be quite interesting but there are some question marks. In my opinion, Authors should get the chance to address the comments, therefore I recommend the major revision. Hope, that my suggestions help to improve the quality of the paper.

Reviewer 2 Report

Teshom et al. study the distribution of EBV genotypes 1 & 2 in lymphoma patients in Ethiopia using a simple PCR test based on size of a product amplified from the EBNA3C gene.  Viral load was also quantified.

Overall, this is a good-sized cohort of 207 patient samples from a both Hodgkin lymphoma and non-Hodgkin lymphoma types. No obvious correlations were made between EBV genotype and/or coinfection and any specific lymphoma or sociodemographic feature.

The obvious question not addressed is connection with clinical outcomes, including survival, recurrence, etc.  If they don't have this data, the authors should comment on why the clinical outcome data is not available.  There are lots of reasons why that may occur, but they should be stated. 

This manuscript is mostly well-written.  A few minor wording corrections might improve it. A short list is provided below:

The figure legends are not indented correctly for MDPI standards

line 47 - I think there are a lot more than several viral miRNAs.

line 59 - use "more often" than "more"

line 106 - don't use the word "particle".  That implies an encapsidated virus. This is genome quantification and is better off stated as genomic copies.

line 125 - "..had EBV genotype 2 or were coinfected with both genotypes respectively."

line 154 - "first" is used too many times in this sentence

line 280 - remove "Please add:"

Author Response

Please the attachment. 

Round 2

Reviewer 1 Report

I would like to thank the Authors for the revised version of manuscript. They addressed most of my comments.

Authors provided more details to the introduction and material and methods sections and that makes the paper more clear.

I have just few small comments.

Introduction

Lines 90-99 - I appreciate that you included information about both Sanger sequencing and NGS. Small suggestion. You are right that Ilumina is vastly used in analysis of polymorphism, metagenomics etc. Nevertheless, there are more options (Nanopore, PacBio), so I would just include next-generation sequencing without mentioning Ilumina as there are more platforms.

Material and methods

This section is far more clear than before (details about deparaffinization, ACTB gene details).

Please add few more details - Asking about elution volume, I meant the volume after extraction, please include this detail.

Lines 322-340 - Express range as copy number or DNA concentration, not the dilution (use notification with order of magnitude – 10n) .

Results and discussion

I have no comments to these sections. The quality of figures was improved, Authors also included supplementary materials answering my question.

However, I am a bit confused with the sequences provided in the supplementary materials. I suppose that there are sequences and primer annealing sites, but you should also provide the caption to it.

Concluding, in my opinion paper will meet the requirements of acceptance as soon as the minor issues are explained.

Hope, I was helpful and looking forward for the further studied concerning the clinical status of EBV patients.  
